# Pharmacokinetic/Pharmacodynamic Analysis of Oral Calcium Fosfomycin: Are Urine Levels Sufficient to Ensure Efficacy for Urinary Tract Infections?

**DOI:** 10.3390/pharmaceutics15041185

**Published:** 2023-04-07

**Authors:** Alicia Rodríguez-Gascón, Ana Alarcia-Lacalle, María Ángeles Solinís, Ana del Pozo-Rodríguez, Zuriñe Abajo, María Cabero, Andrés Canut, Arantxa Isla

**Affiliations:** 1Pharmacokinetic, Nanotechnology and Gene Therapy Group (Pharma Nano Gene), Faculty of Pharmacy, Centro de Investigación Lascaray Ikergunea, University of the Basque Country UPV/EHU, Paseo de la Universidad 7, 01006 Vitoria-Gasteiz, Spain; 2Bioaraba, Microbiology, Infectious Disease, Antimicrobial Agents, and Gene Therapy, 01009 Vitoria-Gasteiz, Spain; 3Bioaraba, Clinical Trials Unit, 01009 Vitoria-Gasteiz, Spain; 4Bioaraba, Microbiology Service, Araba University Hospital, Osakidetza Basque Health Service, 01009 Vitoria-Gasteiz, Spain

**Keywords:** urinary tract infections, calcium fosfomycin, PK/PD analysis, Monte Carlo simulation

## Abstract

Urinary tract infections (UTIs) are extremely common and a major driver for the use of antimicrobials. Calcium fosfomycin is an old antibiotic indicated for the treatment of UTIs; however, data about its urine pharmacokinetic profile are scarce. In this work, we have evaluated the pharmacokinetics of fosfomycin from urine concentrations after oral administration of calcium fosfomycin to healthy women. Moreover, we have assessed, by pharmacokinetic/pharmacodynamic (PK/PD) analysis and Monte Carlo simulations, its effectiveness considering the susceptibility profile of *Escherichia coli*, the main pathogen involved in UTIs. The accumulated fraction of fosfomycin excreted in urine was around 18%, consistent with its low oral bioavailability and its almost exclusively renal clearance by glomerular filtration as unchanged drug. PK/PD breakpoints resulted to be 8, 16, and 32 mg/L for a single dose of 500 mg, a single dose of 1000 mg, and 1000 mg q8h for 3 days, respectively. For empiric treatment, the estimated probability of treatment success was very high (>95%) with the three dose regimens, considering the susceptibility profile of *E. coli* reported by EUCAST. Our results show that oral calcium fosfomycin at a dose level of 1000 mg every 8 h provides urine concentrations sufficient to ensure efficacy for the treatment of UTIs in women.

## 1. Introduction

Urinary tract infections (UTIs) are among the most common infections encountered in the clinic and remain a top indication for women to receive antibiotics. More than one third of women suffer a UTI at some point in their lives and 74% of women who seek treatment for UTI symptoms will receive an antibiotic prescription [1]. Although a wide range of Gram-positive and Gram-negative bacteria may be detected as the causative organisms, *Escherichia coli* is the most frequently isolated microorganism, present in up to 80% of UTIs [2,3]. UTIs are associated with significant disease burden, antimicrobial resistance (AMR), and cost. In 2020, the World Health Organization (WHO) published a target product profile to guide the urgent development of new oral antimicrobial agents for UTIs [4]. As an alternative solution to circumvent the long and costly process of developing new antibiotics, the reassessment and reintroduction of “old” antibiotics have emerged [5], with fosfomycin being one of such “old” antibiotics, developed more than 40 years ago [6].

Fosfomycin is an antimicrobial agent first isolated from *Streptomyces fradiae* and *Pseudomonas syringae*. It exerts bactericidal antimicrobial activity against susceptible microorganisms by blocking the early stage of bacterial cell wall synthesis [7]. This drug, currently produced by a synthetic method, is a low-molecular weight (138 g/mol), highly polar phosphonic acid derivative (cis-1,2-epoxypropyl phosphonic acid) that represents its own class of antibiotics [8,9]. For oral administration, there are two formulations: calcium salt and trometamol (or tromethamine) salt, which differ in their pharmacokinetic properties [10]. Oral bioavailability is greater for the trometamol formulation, whereas the apparent volume of distribution and total body clearance are higher for the calcium salt [11]. Additionally, there is a more hydrophilic salt (fosfomycin disodium) for parenteral administration. Fosfomycin is excreted unchanged in the urine through glomerular filtration [12] and therefore, it is indicated for the treatment of urinary tract infections, including pyelonephritis and cystitis.

Calcium fosfomycin is approved for the treatment of UTIs in women [13]. The dose in adults is 500 mg–1000 mg every 8 h. Up to date, the available information and a positive safety profile of this drug are considered enough to establish a positive benefit–risk balance for this indication [14]. However, information about the fosfomycin urine pharmacokinetic profile after oral administration of the calcium salt is scarce and further studies are needed, including pharmacokinetic and pharmacokinetic/pharmacodynamic (PK/PD) analyses.

The objective of this work was to characterize the urine pharmacokinetics of fosfomycin in healthy women after oral administration of calcium fosfomycin, and to assess if PK/PD analysis supports the effectiveness of the standard treatment considering the susceptibility profile of *E. coli*, the main pathogen involved in UTIs.

## 2. Materials and Methods

### 2.1. Study Design and Subjects

This was a phase I open-label, randomized, crossover study of single and multiple dose of calcium fosfomycin capsules in healthy volunteer women under fasting conditions, carried out at the Clinical Trial Unit, Hospital Universitario Araba (Vitoria-Gasteiz, Spain). The study was performed in accordance with both national and international regulations (ICH Guidelines) applicable to clinical trials, and the principles of the declaration of Helsinki and its subsequent modifications. The study was approved by the Ethics Committee for Investigation with medicinal products of Euskadi (Vitoria-Gasteiz, Spain) and had the authorization of the Spanish Agency of Medicines and Healthcare Products (AEMPS). Written informed consent was obtained from each subject prior to the conduct of any study-related procedure.

Inclusion criteria comprised healthy women, aged between 18 and 55 years, body mass index (BMI) ranging from 18.5 kg/m^2^ to 30 kg/m^2^, no evidence of significant organic or psychiatric disease, clinical laboratory data (including red blood cell count, hemoglobin, hematocrit, serum and urine glucose, cholesterol, triglycerides, urea, serum creatinine, bilirubin, transaminases, alkaline phosphatase, and urine protein, among others) within normal limits, according to normal reference values of the Biochemistry, Hematology and Microbiology Services of the Hospital Universitario Araba, negative test for hepatitis B and C virus and human immunodeficiency virus, normal electrocardiogram and vital signs, effective hormonal and non-hormonal contraceptive, and able to communicate effectively and provide written informed consent. On the one hand, pharmacokinetic studies have not shown any systematic interaction between antibiotics and oral contraceptives steroids [15]. Moreover, no contraindications exist for the administration of fosfomycin with other medications. On the other hand, a low risk for contraceptive failure exists when fosfomycin is co-administered with conjugated estrogens [13]. Exclusion criteria included history of allergy or hypersensitivity to drugs and/or excipients, smoker, positive for drugs of abuse at the time of housing for each experimental visit, usage of any prescribed medication, over-the-counter (OTC) medicinal products, and/or herbal products during the last 14 days preceding the treatment dosing or when the elimination half-life does not ensure their disappearance from the body in good time at the commencement of each experimental visit, breastfeeding and/or pregnancy, major surgery during the previous six months, have donated blood in the twelve weeks prior to the commencement of the trial, and taking part in another clinical trial during the two months prior to the current trial.

### 2.2. Fosfomycin Administration

Every woman received calcium fosfomycin (Fosfocina^®^ 500 mg capsule, Laboratorios ERN, S.A. (Barcelona, Spain)) according to a randomized sequence: single dose of 500 mg (1 capsule of Fosfocina^®^), single dose of 1000 mg (2 capsules of Fosfocina^®^), and three daily doses of 1000 mg (2 capsules of Fosfocina^®^) separated by 8 h for three days. When the women received multiple doses, they received one morning dose (before breakfast), one lunch dose (before a meal), and one evening dose (before dinner); on day 4 (sampling day), they received a new morning 1000 mg dose. All women received the three dose regimens and the order of administration of each treatment (sequence) was determined on the basis of a randomization procedure. A washout period of at least 1 week between treatments was applied. The fosfomycin capsule was administered with 200 mL of water. When subjects received the single dose of calcium fosfomycin, they were fasted for at least 8 h prior to dosing to 4 h post dose. In the case of multiple doses, women were fasted for at least 8 h prior to the last dosing (day 4). During housing days, a standardized diet was delivered to each subject at appropriate time intervals. All meal plans were identical in each study period. In all cases, water was restricted for at least 1 h prior to dosing to 1 h post dose (except for the 200 mL of drinking water given during dosing). At other times, drinking water was provided ad libitum.

In order to assess treatment compliance, when fosfomycin was administered as a single dose, the investigator checked the mouth with the help of a torch light and a tongue depressor after capsule administration. For the multiple-dose regimen, subjects were asked to return unused capsules.

### 2.3. Sample Collection and Fosfomycin Quantification

When fosfomycin was given as single dose (500 mg or 1000 mg), urine samples were collected before and at intervals of 0 to 2, 2 to 4, 4 to 6, 6 to 8, 8 to 12, 12 to 18, 18 to 24, and 24 to 36 h after ingestion. When women received the multiple-dose regimen, urine samples (in the same time intervals as for the single dosing) were collected after the last dose. The volume of each urine sample was recorded and 5 mL aliquots were stored at −80 °C until analysis. The concentration of fosfomycin in the urine samples was measured by Kymos Pharma Services, S.L. (Cerdanyola del Vallès, Spain) with a validated liquid chromatography-tandem mass spectrometry (HPLC-MS/MS) method using an Agilent 1200 chromatography system coupled to a mass spectrometer (API3200, AB Sciex (Framingham, MA, USA)). Linearity was demonstrated from 10 mg/L (lower limit of quantification) to 5000 mg/L (upper limit of quantification). Inter-day and intra-day precision (expressed as the coefficient of variation) of the lower limit of quantification were <10%, and inter-day and intra-day accuracy (expressed as the relative error) of the lower limit of quantification were <15%. Inter-day and intra-day precision of the quality controls (30, 250, 2500, and 4000 mg/L) were <12%, and inter-day and intra-day accuracy of the quality controls were <12%. The matrix effect, expressed as the coefficient of variation, was 6.19% and 4.82% for 30 and 4000 mg/L, respectively. Recovery was 90.08%, 88.69%, and 87.16% for 30, 250, and 4000 mg/L, respectively. Appendix A shows stability data obtained in the validation process.

### 2.4. Pharmacokinetic/Pharmacodynamic (PK/PD) Analysis

Noncompartmental PK analysis was performed on individual urine concentration data using Phoenix 64 (Build 8.3.4.295, Certara USA, Inc., Princeton, NJ, USA). The urine PK parameters calculated included urine excretion rate, urine half-life (t_1/2_), amount of drug excreted in urine (Ae), percentage of dose excreted unchanged in urine (fe), and maximum excretion rate. Additionally, the area under the urine concentration curve in a period of 24 h (AUC_urine,24_) was calculated with the trapezoidal rule by assuming that the drug concentration remains constant in the urinary bladder during each time interval [16].

Fosfomycin exhibits concentration-dependent killing activity against *E. coli* and the AUC/minimum inhibitory concentration (MIC) ratio of 24 has been associated with efficacy [17,18]. Therefore, for PK/PD analysis, Monte Carlo simulations based on the AUC_urine,24_ distribution were carried out to estimate the probability of reaching the AUC_urine,24_/MIC ratio of 24. Monte Carlo simulations (10,000 subjects) were conducted using Oracle^®^Crystal Ball Fusion Edition v.11.1.2.3.500 (Oracle USA Inc., Redwood City, CA, USA). For simulations, a log-normal distribution of AUC_urine,24_ was assumed. The probability of target attainment or PTA, defined as the probability that a specific value of a PK/PD index associated with the efficacy of the antimicrobial treatment is achieved at a certain MIC [19], was estimated for a MIC range from 0.125 to 516 mg/L. For every MIC value and considering the variability of the AUC_urine,24_, the output of the simulation consisted of a probability distribution, and the mean, median, and 95% confidence interval (CI) (expressed as percentiles) of the AUC_urine,24_/MIC were extracted. The PK/PD breakpoints, considered as the highest MIC value at which AUC_urine,24_/MIC is ≥24, were estimated according to EUCAST; that is, the PK/PD breakpoint was obtained from the lower limit of the 95% CI (2.5% percentile) [20].

Cumulative fraction of response (CFR) values, which allow us to calculate the probability of success of an empiric treatment, were calculated considering the PTA for each MIC value and the bacterial population MIC distribution. Susceptibility data for *E. coli* reported by EUCAST were used [21].

PTA and CFR ≥80% but <90% are associated with moderate probabilities of success, whereas PTA and CFR ≥90% are considered as optimal [22].

### 2.5. Statistical Analysis

Urine half-lives of fosfomycin obtained from the subjects when they received different doses were compared using paired *t*-test in the IBM^®^ SPSS^®^ Statistics software (V. 27). Differences were considered significant if *p* < 0.05.

## 3. Results

A total of 24 healthy women were enrolled in the study. Table 1 shows the characteristics of the subjects. For the multiple-dose regimen, two subjects did not complete the dosing schedule properly and another two did not complete the sampling schedule; therefore, for this dose level, these four patients were excluded.

Figure 1 shows the percentage of administered dose recovered in urine over time following oral administration of a single dose of 500 mg or 1000 mg. Of the total dose administered, the cumulative fraction excreted was around 18%, regardless of the dose level.

Table 2 shows the urine pharmacokinetic parameters. The urine half-life was around 7 h and no significant difference was detected depending on the dose.

Figure 2 and Figure 3 show the individual and the mean fosfomycin concentrations in the urine samples in comparison with MIC values of 8, 16, and 32 mg/L. In either single or multiple dosing, urine concentration was higher than 8 mg/L (the EUCAST clinical breakpoint of fosfomycin for *E. coli* in UTIs) for at least 24 h in almost all samples. When calcium fosfomycin was administered for three days (3 g daily), in most patients, concentrations were higher than 32 mg/L for at least 24 h.

Table 3 shows the maximum concentration (C_max_) of fosfomycin in the urine samples, as well as the C_max_ to MIC ratios for MICs of 4 and 8 mg/L, the tentative epidemiological cut-off (TECOFF) and the clinical breakpoint for *E. coli* reported by EUCAST, respectively. After administration of the 500 mg single dose, the C_max_ was 33 and 16 times higher than the TECOFF and the clinical breakpoint for *E. coli*, respectively. When the women received 1000 g q8 h for 3 days, these C_max_ to MIC ratios reached values of 73 and 37, respectively.

Figure 4 shows the PTA for the three dose levels of fosfomycin and the MIC distribution of *E. coli* against fosfomycin reported by EUCAST. On the basis of simulation results and taking into account the PK/PD target, probabilities of AUC_0–24h_/MIC >24 higher than 90% are achieved for MIC values equal to or lower than 16 mg/L for a 500 and 100 mg single dose, and 64 mg/L for 1000 mg q8h for three days.

Figure 5 features the relationship between AUC_urine,24_/MIC and MIC for the three different dosing regimens of fosfomycin. The PK/PD breakpoint, which is the highest MIC value at which there is a high probability of target attainment, can be read directly from the figure at the intersection of the horizontal line at the PK/PD target and the lower limit of the 95% confidence interval (2.5% percentile). According to the obtained results, the MIC values supposedly covered are 8, 16, and 32 mg/L for the single dose of 500 mg, the single dose of 1000 mg, and 1000 mg q8h for 3 days, respectively.

Table 4 shows the AUC_urine,24_/MIC distribution (mean, median, and 2.5% percentile) and the CFR value of fosfomycin against *E. coli* considering the MIC distribution reported by EUCAST. Irrespective of the dose level, CFR values were ≥90%, which is associated with a high probability of treatment success.

## 4. Discussion

Data available about the urine fosfomycin pharmacokinetic profile for the calcium formulation after oral administration are scarce. Available concentrations in urine are often extrapolated from data published for fosfomycin trometamol and need to be interpreted with caution. In this work, we have characterized the urine pharmacokinetic profile of fosfomycin after oral administration to 24 healthy women at different dose levels of the calcium salt. Moreover, we applied PK/PD analysis to compare the urine concentrations with susceptibility data of *E. coli*, the main pathogen involved in UTIs.

There are several sites of infection that, for different reasons, are exposed to drug concentrations that differ from what would be predicted from those in plasma or serum [23]. Among those are infections occurring within the urinary tract. Actually, and according to the WHO [4], pharmacokinetic data (renal elimination) and activity in urine must support oral therapy of UTIs. Therefore, for this indication, it is the unbound drug concentrations within urine versus the MICs in urine that should be considered [24].

The cumulative fraction excreted in urine 36 h after fosfomycin administration was around 18%. This value is similar to that reported in a previous study in which the pharmacokinetics of fosfomycin were evaluated after oral administration of calcium fosfomycin to human volunteers [25]. This excretion rate is consistent with the low oral bioavailability (20–30%) due to hydrolysis by gastric acid [26], and with its almost exclusively renal clearance by glomerular filtration as unchanged drug [27]. Fosfomycin urine half-life (around 7 h) was similar to previously reported [28].

Although PK/PD analysis may be considered less useful for the treatment of UTIs because the antibiotics can often reach relatively high urinary concentrations (much higher than the MICs of the bacteria responsible for the infection), patient’s conditions (pH of the urine, risk factors such as pre-existing diseases, reduced oral absorption, faster elimination…) and bacterial strain characteristics (lower susceptibility, virulence factors, etc.) may play an important role [29]. In fact, in our study, urine concentrations presented high variability. In a previous study in which fosfomycin trometamol was administered to healthy female volunteers, a high between-subject variability was also found [30]. This variability might explain treatment failures. Since PK/PD analysis with Monte Carlo simulations considers the inter-patient variability that provides information on the influence of patient characteristics on the PK behavior of the drug, it is a very useful tool to establish rational dosage regimens of antimicrobial agents in human and veterinary medicine [31]. The use of PK/PD analyses can ameliorate the risk of selecting multidrug-resistant isolates and improve the likelihood of selecting an effective dose regimen, thereby increasing the likelihood of success. These tools have also been applied to identify changes in the antimicrobial activity of antibiotics, providing complementary information to the simple assessment of MIC values [32,33,34], and to establish PK/PD breakpoints [35] as well.

There is confusion in the literature about whether fosfomycin displays time- or concentration-dependent bactericidal activity. Roussos et al. [12] have reported that the activity profile may be dependent on the microorganism. Fosfomycin exhibits concentration-dependent killing activity against strains of *E. coli*, *Proteus mirabilis*, and *Streptococcus pneumonie*, and time-dependent bactericidal activity against *Staphylococcus aureus* and *Pseudomonas aeruginosa* [12,36]. In a recent study, Lepak et al. [18] demonstrated that the AUC/MIC ratio is the PK/PD index most closely linked to efficacy in a murine model. These authors also observed that for *E. coli*, the maximum survival (100%) occurred at AUC/MIC ratios of 24, which encompasses their identified stasis target. This PK/PD target has been applied in a study to evaluate fosfomycin against *E. coli* and *Klebsiella* spp. from urinary tract infections [17].

According to the PTA values, a high probability of target success is achieved for isolates with MIC ≤16 mg/L with a single dose of either 500 or 1000 mg, and ≤64 mg/L with 1000 mg q8h for three days. We have also calculated the PK/PD breakpoint of fosfomycin based on the relationship of the PK/PD index (AUC_urine,24_/MIC) as a function of the MIC [20], which provides a much more restrictive PK/PD breakpoint than considering the PTA >90%. Applying this criterion, the PK/PD breakpoint of fosfomycin resulted to be 8 mg/L, 16 mg/L, and 32 mg/L for a single dose of 500 mg, a single dose of 1000 mg, and 3 g daily for three days, respectively. Fosfomycin has a wide spectrum of in vitro activity against Gram-negative bacteria, especially *E. coli* [37]. In 2021, EUCAST changed the oral fosfomycin clinical breakpoints from being for all urinary *Enterobacterales* (S ≤ 32 mg/L; R > 32 mg/L) to being for *E. coli* only and lowered the breakpoint (S ≤ 8 mg/L; R > 8 mg/L). Additionally, EUCAST has established the tentative epidemiological cut-off for *E. coli* at 4 mg/L [38]. Therefore, only for the 500 mg single dose, the PK/PD breakpoint matched the EUCAST clinical breakpoint for *E. coli* (8 mg/L) [21]. Regardless of the dose, the fosfomycin PK/PD breakpoints were higher than the epidemiologic cut-off value ECOFF, that is, the highest MIC for organisms devoid of phenotypically detectable acquired resistance mechanisms.

Although 500 and 1000 mg single doses provide a high probability of treatment success, given the large between-subject variability in the urine concentrations of fosfomycin, a multiple-dose regimen may be of higher clinical benefit, especially for those patients with low urine concentrations. Although EUCAST establishes a PK/PD breakpoint of oral fosfomycin for uncomplicated UTIs (8 mg/L), it does not differentiate between calcium or trometamol fosfomycin [38].

CFR is an index that estimates the probability of target attainment for an MIC distribution, and therefore it is very useful to predict the probability of treatment success of an antimicrobial agent when applied empirically [39]. Considering the MIC distribution reported by EUCAST [21], which includes 11 different distributions and 2351 observations, calcium fosfomycin provides a high (>95%) probability of treatment success.

According to our PK/PD study, concentrations measured in the urine of women confirm that calcium fosfomycin given orally (1000 mg every 8 h) are sufficient to ensure efficacy for urinary tract infections. To the best of the authors’ knowledge, this is the first study describing the urine pharmacokinetic profile in women receiving calcium fosfomycin, including a PK/PD analysis and Monte Carlo simulations.

A limitation of this work is that healthy women instead of women with urinary tract infections were included. Contrary to other studies in which the volunteers collected the samples based on their own urinary rhythm [30], urine sampling in our study was scheduled to ensure a higher number of samples (nine in 36 h); however, we regard this as a strength since UTIs result in a high amount of small urine portions [30]. Another limitation of this work is the high variability of fosfomycin in the urine samples. As mentioned above, drug concentrations in urine depend on the dose and dose regimen, drug formulation, PK behavior, rate and extension of urine formation (affected by fluid intake), and voiding frequency; all of these factors are responsible for high fluctuations [24]. In our study, we included young healthy subjects without renal insufficiency, and no volunteer was obese (BMI < 30 kg/m^2^). During the sampling period, water was restricted from at least 1 h prior to dosing to 1 h post dose (except for the 200 mL of drinking water given during dosing), and the volunteers received a standardized diet (from 4 h after fosfomycin administration), the same for each subject. Despite these conditions and the scheduled sampling, the variability was still high, although lower than expected in clinical practice, where fluctuations in urine concentrations, along with the corresponding within- and between-patient variability, can be extremely large, magnifying the challenges associated with using urinary drug concentrations as the biophase component of a PK/PD assessment. Another limitation of our study is the number of subjects included (24 women), lower than in a previous study carried out by Wijma et al. [28] in which the interindividual variability in urinary fosfomycin concentrations in 40 healthy female volunteers was evaluated. Other studies have included less than 24 subjects. For instance, Goto et al. [25] evaluated the pharmacokinetics of fosfomycin administered by intravenous and oral route in only seven volunteers, and Matsumoto et al. [40] studied the pharmacokinetics of calcium fosfomycin in 20 healthy individuals.

Our results are in agreement with a previous study in which the clinical effect of a two-day treatment with calcium fosfomycin for acute uncomplicated cystitis in women was evaluated [40]. In that study, overall evaluation of the cure revealed that microbiological eradication rate (microbiological outcome) and clinical efficacy rate (clinical outcome) at 5–9 days after drug administration were 94.9%.

## 5. Conclusions

Our study confirms the utility of PK/PD analysis and Monte Carlo simulations to predict the efficacy of antibiotics excreted in urine and used for the treatment of UTIs. One advantage of this kind of studies is the possibility of comparing different dose regimens.

The PK/PD breakpoints of calcium fosfomycin estimated from urine pharmacokinetic data resulted to be 8 mg/L, 16 mg/L, and 32 mg/L for a single dose of 500 mg, a single dose of 1000 mg, and 3 g daily for three days, respectively. Only for the lowest dose level, the PK/PD breakpoint matches the clinical breakpoint of EUCAST for *E. coli*. For empiric treatment, and considering the MIC distribution reported by EUCAST, calcium fosfomycin provides a high (>95%) probability of treatment success.

Our results show that oral calcium fosfomycin at a dose level of 1000 mg every 8 h for three days provides urine concentrations sufficient to ensure efficacy for the treatment of UTIs in women. However, the results obtained in this study carried out with healthy women and based on urine PK/PD analysis must be correlated with well-designed efficacy studies in women with UTIs.

## Figures and Tables

**Figure 1 pharmaceutics-15-01185-f001:**
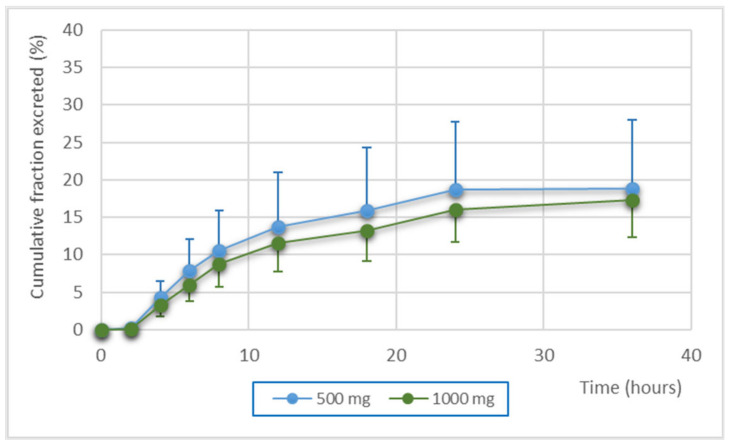
Cumulative fraction of administered dose recovered in urine (mean and standard deviation) after a single administration of 500 mg or 1000 mg.

**Figure 2 pharmaceutics-15-01185-f002:**
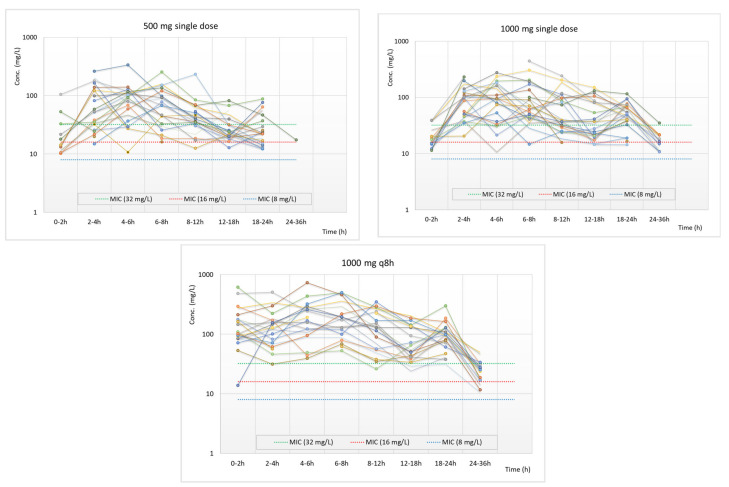
Individual concentrations of fosfomycin in the urine of the subjects (each color represents a different subject) over time in comparison to MIC values of 8, 16, and 32 mg/L (blue, red, and green lines, respectively). For 1000 mg given every 8 h, sampling starts after the administration of the last dose.

**Figure 3 pharmaceutics-15-01185-f003:**
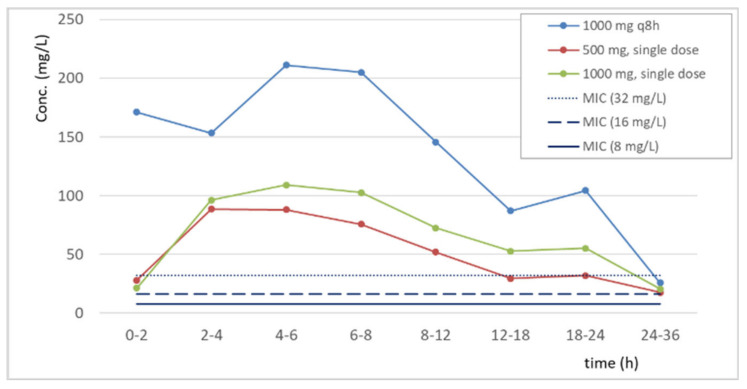
Mean concentrations of fosfomycin in the urine samples in comparison to MIC values of 8, 16, and 32 mg/L. For 1000 mg given every 8 h, sampling starts after the administration of the last dose.

**Figure 4 pharmaceutics-15-01185-f004:**
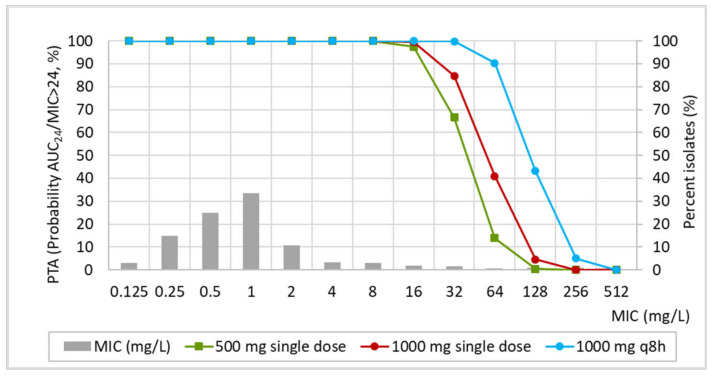
Probability of target attainment (PTA) for fosfomycin and MIC distribution of *E. coli* against fosfomycin reported by EUCAST.

**Figure 5 pharmaceutics-15-01185-f005:**
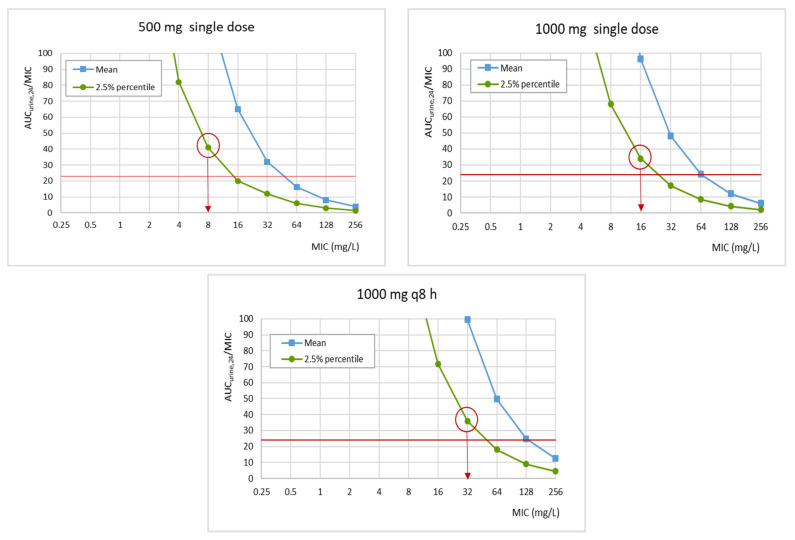
Relationship between AUC_urine,24_/MIC and MIC for fosfomycin.

**Table 1 pharmaceutics-15-01185-t001:** Characteristics of the healthy women who participated in the study.

	Age (years)	Body Weight (kg)	Height (cm)	BMI (kg/m^2^)	CrCL (mL/min)
Mean ± SD	32 ± 9	65 ± 10	165 ± 6	23.89 ± 2.53	109 ± 21
Minimum	19	52	155	20.00	83
Maximum	49	95	178	29.90	158

BMI: body mass index; CrCL: creatinine clearance estimated by Cockroft-Gault from serum creatinine concentration; SD: standard deviation.

**Table 2 pharmaceutics-15-01185-t002:** Urine pharmacokinetic parameters of fosfomycin after oral administration to the healthy women. Data are expressed as mean ± standard deviation.

Dose	t_1/2_ (h)	Amount Recovered (mg)	Cumulative Fraction Excreted (%)	Maximum Excretion Rate (mg/L)
500 mg single dose (n: 24)	6.89 ± 3.10	95.94 ± 45.56	18.79 ± 9.17	12.19 ± 6.51
1000 mg single dose (n: 24)	6.88 ± 2.56	174.25 ± 51.40	17.35 ± 4.95	20.38 ± 9.47
1000 mg q8h (n: 20)	7.97 ± 2.11	-	-	38.75 ± 13.69

**Table 3 pharmaceutics-15-01185-t003:** C_max_ and C_max_/MIC ratio of urine fosfomycin in the volunteers receiving fosfomycin at different dose levels. Values expressed as mean ± standard deviation. In parenthesis, maximum and minimum values.

	C_max_ (mg/L)	C_max_/4	C_max_/8
500 mg single dose	133.72 ± 80.34 (21.79–336.87)	33.43 ± 20.08 (5.45–84.22)	16.71 ± 10.04 (2.72–42.11)
1000 mg single dose	158.20 ± 90.62 46.09–443.07)	39.55 ± 22.65 (11.52–110.77)	19.78 ± 11.33 (5.76–55.38)
1000 mg q8h, 3 days	293.54 ± 175.22 (77.90–731.10)	73.38 ± 43.80 (19.48–182.78)	36.69 ± 21.90 (9.74–91.39)

**Table 4 pharmaceutics-15-01185-t004:** AUC_urine,24_/MIC distribution (mean, median, and 2.5% percentile) and CFR values.

	CFR (%)	2.5% Percentile	Median	Mean
500 mg single dose	97	16	1166	1816
1000 mg single dose	98	24	1743	2711
1000 mg q8h, 3 days	98	54	3516	5515

## Data Availability

The data are not publicly available due to privacy or ethical restrictions.

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
