# Peer review of "Pharmacokinetic/Pharmacodynamic Analysis of Oral Calcium Fosfomycin: Are Urine Levels Sufficient to Ensure Efficacy for Urinary Tract Infections?"

_pharmaceutics, 2023, doi:10.3390/pharmaceutics15041185_

Round 1

Reviewer 1 Report

Line 56. Change “unmetabolized” to “unchanged”.

Lines 59-65. I don’t clearly understand the rationale for this study. At the end of the Introduction, you say that calcium fosfomycin is approved for UTIs in women, but then say that further studies are needed to confirm an appropriate dose and efficacy. Has this not already been determined from the studies used to support registration for this indication? It’s only in the first paragraph of the discussion (lines 235-241) that I get a clearer picture of why the study is being done – this info should be in the Introduction!

Line 87. Please clarify that none of the hormonal contraceptive measures used by participants in this study are likely to alter the PK or fosfomycin.

Line 133. Should read “area under the urine concentration curve”.

Lines 140-141. How confident are you in the precision dosing (PK/PD) target of AUCurine,24/MIC = 24? If this has already been determined, then there must be a reasonably good understanding of the PK/PD of fosfomycin for UTIs, so the exact rationale for this study is unclear. Again, it is only in the discussion that this is well explained (lines 280-283).

Line 166. Replace “other” with “another”.

Figure 1. Is the accumulated fraction excreted calculated by oral dose (mg) / total amount in urine (mg)? Thus, for the 1000 mg dose, about 180 mg is collected in the urine, to give about 18%? This value is therefore different from the fraction excreted unchanged (fe), which is defined as renal clearance/total clearance and independent of bioavailability i.e., you can only clear a drug that has entered the body (total clearance = F x Dose / AUC). Please clarify.

Line 179. What is “urine plasma”?

Conclusion. Is there a stronger conclusion that currently written? One that more broadly speaks to the value of this type of modelling and simulation and having a precision dosing target? There are already efficacy studies establishing the indication. What is of interest would be to use M&S to predict those women who may require a different dose regimen than the approved doses.

Author Response

Reviewer 1

Comments and Suggestions for Authors

Line 56. Change “unmetabolized” to “unchanged”.

“Unmetabolized” has been replaced by “unchanged”.

Lines 59-65. I don’t clearly understand the rationale for this study. At the end of the Introduction, you say that calcium fosfomycin is approved for UTIs in women, but then say that further studies are needed to confirm an appropriate dose and efficacy. Has this not already been determined from the studies used to support registration for this indication? It’s only in the first paragraph of the discussion (lines 235-241) that I get a clearer picture of why the study is being done – this info should be in the Introduction!

We thank this comment from the reviewer. We have modified the introduction section accordingly. In the new version of the manuscript, this paragraph in the introduction section is as follow:

Calcium fosfomycin is approved for the treatment of UTIs in women [13]. The dose in adults is 500 mg-1000 mg every 8 hours. Up to date, the available information and a positive safety profile of this drug are considered enough to stablish a positive benefit-risk balance for this indication [14]. However, information of the fosfomycin urine pharmacokinetic profile after oral administration of the calcium salt is scarce, and further studies are needed, including pharmacokinetic and pharmacokinet-ic/pharmacodynamic (PK/PD) analysis.

Line 87. Please clarify that none of the hormonal contraceptive measures used by participants in this study are likely to alter the PK or fosfomycin.

We have added the following information to clarify this point (lines 91-95):

On the one hand, pharmacokinetic studies have not shown any systematic interaction between antibiotics and oral contraceptives steroids [15]. Moreover, no contraindications exist for the administration of fosfomycin with other medications. On the other hand, a low risk for contraceptive failure exists when fosfomycin is coadministered with conjugated estrogens [13].

Line 133. Should read “area under the urine concentration curve”.

We have replaced “area under de urine concentration versus time curve”. by “area under the urine concentration curve”.

Lines 140-141. How confident are you in the precision dosing (PK/PD) target of AUCurine,24/MIC = 24? If this has already been determined, then there must be a reasonably good understanding of the PK/PD of fosfomycin for UTIs, so the exact rationale for this study is unclear. Again, it is only in the discussion that this is well explained (lines 280-283).

We have modified lines 140-141 to clarify this point. In the new version of the manuscript, this paragraph is as follow (lines 156-157):

“Fosfomycin exhibits concentration-dependent killing activity against E. coli, and an AUC/MIC ratio of 24 has been associated with efficacy [15,16]. Therefore, for PK/PD analysis,…”

Line 166. Replace “other” with “another”.

We have replaced “other” with “another”.

Figure 1. Is the accumulated fraction excreted calculated by oral dose (mg) / total amount in urine (mg)? Thus, for the 1000 mg dose, about 180 mg is collected in the urine, to give about 18%? This value is therefore different from the fraction excreted unchanged (fe), which is defined as renal clearance/total clearance and independent of bioavailability i.e., you can only clear a drug that has entered the body (total clearance = F x Dose / AUC). Please clarify.

Cumulative fraction excreted was calculated over the total dose administered, and therefore, it depends on the oral bioavailability and the unchanged fraction excreted. To better clarify this point, we have modified the paragraph and it is now as follows (lines 193-195):

Figure 1 shows the percentage of administered dose recovered in urine over time following oral administration of single dose of 500 mg or 1000 mg. Of the total dose administered, the cumulative fraction excreted was around 18%, regardless of the dose level.

Line 179. What is “urine plasma”?

We apologize for the mistake. It should be “urine half-life”. We have corrected it.

Conclusion. Is there a stronger conclusion that currently written? One that more broadly speaks to the value of this type of modelling and simulation and having a precision dosing target? There are already efficacy studies establishing the indication. What is of interest would be to use M&S to predict those women who may require a different dose regimen than the approved doses.

We thank the reviewer for this comment. We have modified the conclusion section. In the new version of the manuscript, it is as follows:

Our study confirms the utility of PK/PD and Monte Carlo simulations to predict the efficacy of antibiotics excreted in urine and used for the treatment of UTI. One ad-vantage of this kind of studies is the possibility of comparing different dose regimens.

The PK/PD breakpoints of calcium fosfomycin estimated from urine pharmacokinetic data resulted to be 8 mg/L, 16 mg/L and 32 mg/L for single dose of 500 mg, single dose of 1000 mg, and 3 g daily for three days, respectively. Only for the lowest dose level, the PK/PD breakpoint matches the clinical breakpoint of EUCAST for E. coli. For empiric treatment, and considering the MIC distribution reported by EUCAST, calcium fosfomycin provides a high (>95%) probability of treatment success.

Our results show that oral calcium fosfomycin at a dose level of 1000 mg every 8 h for three days provides urine concentrations sufficient to ensure efficacy for the treatment of UTIs in women. However, the results obtained in this study carried out with healthy women and based on urine PK/PD analysis must be correlated with well-designed efficacy studies in women with UTIs.

Reviewer 2 Report

The manuscript deals with an investigation on urine pharmacokinetics of fosfomycin in healthy women after oral administration of calcium fosfomycin, performing a PK/PD study considering the susceptibility profile of E. coli in UTIs.

This is an important issue in the present context considering the scarce data, lack of agreement in some items, and the need for new approaches, including by reintroduction of the old drugs, to fight against antimicrobial resistances.

However, I found some critical issues that should be addressed:

Q1 - Inclusion/exclusion criteria. Line 85. Regarding Inclusion/exclusion criteria “clinical laboratory values within normal limits” are mentioned. Please clarify this sentence.

Q2 - Fosfomycin is excreted in the urine through glomerular filtration. Thus, renal impairment may influence data results. Considering this, renal function of volunteers should be characterized. Estimated glomerular filtration rate (eGFR) data was not considered. Please clarify.

Q3 - How was rolled out the non-adherence?

Q4 - Regarding fosfomycin quantification (Line 115), a HPLC-MS/MS technique was shortly described, presenting precision and accuracy data. However, validation of analytical technique in PK studies are essential. Considering that the analytical technique was not previously published, a more comprehensive description would be advisable. For instance, no recovery data was presented and no equipment details were added.

Q5 - Stability of the samples is of major importance considering the proposed study design. Thus, more detailed information about the procedure and storage would be advisable. Analytical technique validation should include information about stability of samples.

Q6 - Pharmacokinetic study Table 2 – a more complete data analysis required with variability included.

Q7 - High interindividual variability was stated as a limitation of this work and influencing factor indicated. However, the influence of these volunteer characteristics (e. g. urinary output, number of urinations, estimated glomerular filtration rate (eGFR), BMI, urinary pH) was not discussed.

Q8 - Figure 1. Legend should be completed, including dispersion data.

Q9 – Figure 2 and 3. MIC lines are not highlighted in graph or commented in legend. That would help the reader.

Q10 - The question of the title was not addressed in discussion. Was it? Are these results able to be applied to patients?

Q11 - Both strengths and limitations (including the number of subjects and its statistical power) should be improved/justified.

Q12 - Conclusions should be improved/clarified.

Author Response

Reviewer 2

The manuscript deals with an investigation on urine pharmacokinetics of fosfomycin in healthy women after oral administration of calcium fosfomycin, performing a PK/PD study considering the susceptibility profile of E. coli in UTIs.

This is an important issue in the present context considering the scarce data, lack of agreement in some items, and the need for new approaches, including by reintroduction of the old drugs, to fight against antimicrobial resistances.

However, I found some critical issues that should be addressed:

Q1 - Inclusion/exclusion criteria. Line 85. Regarding Inclusion/exclusion criteria “clinical laboratory values within normal limits” are mentioned. Please clarify this sentence.

To better clarify this point, we have include the following information:

“clinical laboratory data (including red blood cell count, hemoglobin, hematocrit, serum and urine glucose, cholesterol, triglycerides, urea, serum creatinine, bilirubin, transaminases, alkaline phosphatase and urine protein, among others) within normal limits, according to normal reference values of the Biochemistry, Hematology and Microbiology Services of the Hospital Universitario Araba”

Q2 - Fosfomycin is excreted in the urine through glomerular filtration. Thus, renal impairment may influence data results. Considering this, renal function of volunteers should be characterized. Estimated glomerular filtration rate (eGFR) data was not considered. Please clarify.

The study was carried out in healthy women with normal renal function. We included the value of CrCL in table 1 (in the new version table 2). Mean value was 109 mL/min with minimum and maximum of 83 and 158 mL/min.

Q3 - How was rolled out the non-adherence?

When fosfomycin was administered as single dose, after capsule administration, the investigator checked the mouth with the help of torch light and tongue depressor. For multiple dose regimen, the subject's compliance with the treatment was established by maintaining appropriate records of dispensing, return and counting of study treatments. Subjects were asked to return unused capsules.

We have added the following paragraph at the end of 2.2. Fosfomycin administration section (lines 123-126).

“In order to assess treatment compliance, when fosfomycin was administered as single dose, the investigator checked the mouth with the help of torch light and tongue depressor after capsule administration. For multiple dose regimen, subjects were asked to return unused capsules”.

Q4 - Regarding fosfomycin quantification (Line 115), a HPLC-MS/MS technique was shortly described, presenting precision and accuracy data. However, validation of analytical technique in PK studies are essential. Considering that the analytical technique was not previously published, a more comprehensive description would be advisable. For instance, no recovery data was presented and no equipment details were added.

2.3. Sample collection and fosfomycin quantification section has been implemented with more information about the analytical method.

Q5 - Stability of the samples is of major importance considering the proposed study design. Thus, more detailed information about the procedure and storage would be advisable. Analytical technique validation should include information about stability of samples.

Stability of the samples has been included in a table (in the new version, table 1).

Q6 - Pharmacokinetic study Table 2 – a more complete data analysis required with variability included.

Table 2 has been implemented (in the new version table 3). In the new version, the PK analysis is now as follows (lines 149-155):

Noncompartmental PK analysis was performed on individual urine concentration data, using Phoenix 64 (Build 8.3.4.295, Certara USA, Inc., Princeton, NJ, USA). Urine PK parameters calculated included urine excretion rate, urine half-life (t1/2), amount of drug excreted in urine (Ae), percentage of dose excreted unchanged in urine (fe), and maximum excretion rate. Additionally, the area under de urine concentration curve in a period of 24 hours (AUCurine,24) was calculated with the trapezoidal rule by assuming that the drug concentration remains constant in the urinary bladder during each time interval [16].

Q7 - High interindividual variability was stated as a limitation of this work and influencing factor indicated. However, the influence of these volunteer characteristics (e. g. urinary output, number of urinations, estimated glomerular filtration rate (eGFR), BMI, urinary pH) was not discussed.

We appreciate this comment from the reviewer. We have implemented this paragraph (lines 351-358):

“In our study, we included young healthy subjects without renal insufficiency, and no volunteer was obese (BMI < 30 Kg/m2). During the sample period, water was restricted from least 1 hour prior to dosing to 1 hour post-dose (except 200 mL of drinking water given during dosing), and the volunteers received a standardized diet (from 4 hour after fosfomycin administration), the same for each subject. In spite of these conditions and the scheduled sampling, variability is still high, although it is expected to be lower than that in routine clinical practice…”

Q8 - Figure 1. Legend should be completed, including dispersion data.

The legend has been modified according to this comment. Now it is as follow:

“Figure 1. Cumulative fraction of administered dose recovered in urine (mean and standard deviation) after single administration of 500 mg or 1000 mg”

Q9 – Figure 2 and 3. MIC lines are not highlighted in graph or commented in legend. That would help the reader.

This information is in the figure legends and also in the text.

Q10 - The question of the title was not addressed in discussion. Was it? Are these results able to be applied to patients?

We thank this comment of the reviewer. We have added the following comment (lines 338-342):

“According to our PK/PD study, concentrations measured in the urine of women confirm that calcium fosfomycin given orally (1000 mg every 8 hours) are sufficient to ensure efficacy for urinary tract infections. To the best of the author's knowledge, this is the first study describing the urine pharmacokinetic profile in women receiving cal-cium fosfomycin, including PK/PD analysis and Monte Carlo simulations”.

Q11 - Both strengths and limitations (including the number of subjects and its statistical power) should be improved/justified.

Limitations and strengths of the study have been implemented. Since the objective was to obtain pharmacokinetic data, a formal statistical sample size calculation was not performed. However, we have included the sample size as a limitation of the study (lines 360-366).

“Another limitation of our study is the number of subjects included (24 women), lower than in a previous study carried out by Wijma et al [28] in which the interindividual variability in urinary fosfomycin concentrations in 40 healthy female volunteers was evaluated. Other studies included even less than 24 subjects. For instance, Goto et al [41] evaluated the pharmacokinetics of fosfomycin administered intravenously and orally in only 7 seven volunteers, and Matsumoto et al [42] who studied the pharmacokinetics of calcium fosfomycin in 20 healthy individuals”.

Q12 - Conclusions should be improved/clarified.

We thank the reviewer for this comment. We have modified the conclusion section. In the new version of the manuscript, it is as follows:

Our study confirms the utility of PK/PD and Monte Carlo simulations to predict the efficacy of antibiotics excreted in urine and used for the treatment of UTI. One advantage of this kind of studies is the possibility of comparing different dose regimens.

The PK/PD breakpoints of calcium fosfomycin estimated from urine pharmacokinetic data resulted to be 8 mg/L, 16 mg/L and 32 mg/L for single dose of 500 mg, single dose of 1000 mg, and 3 g daily for three days, respectively. Only for the lowest dose level, the PK/PD breakpoint matches the clinical breakpoint of EUCAST for E. coli. For empiric treatment, and considering the MIC distribution reported by EUCAST, calcium fosfomycin provides a high (>95%) probability of treatment success.

Our results show that oral calcium fosfomycin at a dose level of 1000 mg every 8 h for three days provides urine concentrations sufficient to ensure efficacy for the treatment of UTIs in women. However, the results obtained in this study carried out with healthy women and based on urine PK/PD analysis must be correlated with well-designed efficacy studies in women with UTIs.

Reviewer 3 Report

This article regarding the PK/PD analysis of calcium fosfomycin is interesting, it takes into account an adequate number of healthy women volunteer, and gives a clear idea of the behaviour of calcium phosphomycin in the body.

Some clarifications: Is there no interaction of calcium fosfomyn with food?with food?

LInes 118-119: What was the change in the volume of urine collected by the different volunteers? It was an "important" variation?

To lighten the paper a bit I think that figure 2 can be removed, keeping only figure 3 adding the SD.

I appreciate your discussion, it s well done and including a large number of topics of discussion, also the limitations of the work. 

And now you must use calcium fosfomycin on patients with urinary infections and so we can see if theory can be applied to practice!

Author Response

This article regarding the PK/PD analysis of calcium fosfomycin is interesting, it takes into account an adequate number of healthy women volunteer, and gives a clear idea of the behaviour of calcium phosphomycin in the body.

Some clarifications: Is there no interaction of calcium fosfomyn with food?

Evidence suggests than coadministration of fosfomycin with food may reduce absorption (Falagas ME, et al. Clin Microbiol Rev. 2016 Apr;29(2):321-47. doi: 388 10.1128/CMR.00068-15). In our study, women received fosfomycin in fasted state; therefore, interaction with food was avoided.

Lines 118-119: What was the change in the volume of urine collected by the different volunteers? It was an "important" variation?

Although differences in the urine volume were observed between subjects, within subject variability was smaller. Differences may be due, among others, to water intake, that was ad libitum from 1 hour after drug administration. Since the purpose of the study was not to identify variability sources, this factor was not considered.

To lighten the paper a bit I think that figure 2 can be removed, keeping only figure 3 adding the SD.

We appreciate this comment of the reviewer. Actually, we considered this option; however, since the standard deviations are high, the graphic was not very clear. In our opinion, the presentation of mean and individual profiles in independent plots are much more illustrative.

I appreciate your discussion, it s well done and including a large number of topics of discussion, also the limitations of the work.

We appreciate this comment of the reviewer very much.

And now you must use calcium fosfomycin on patients with urinary infections and so we can see if theory can be applied to practice!

Reviewer 4 Report

my concern about this manuscript "Pharmacokinetic/pharmacodynamic analysis of oral calcium 2 fosfomycin: are urine levels sufficient to ensure efficacy for uri- 3 nary tract infections?" is given below-

calcium fosfomycin, calcium can effect the efficiency of medicine  or important for stability of antibiotic?

Abstract: This section is okay but in the last  conclusion, urine concentrations and PK/PD analysis and Monte Carlo simulations sup- 28 port calcium fosfomycin for the treatment of UTIs in women. is looking incomplete, author should provide how much amount of dose is efficient or significant for patient 

Introduction: 

The dose 59 in adults is 500 mg-1000 mg every 8 hours. Up to date, the available information and a 60 positive safety profile of this drug are considered enough to stablish a positive benefit- 61 risk balance for this indication (reference required).

Results and Discussion; Both Section Okay

Image: Author should remove the inside lines of all graph make more clear image  of all image 

Conclusion: Author should more elaborate it 

Author Response

Reviewer 4

My concern about this manuscript "Pharmacokinetic/pharmacodynamic analysis of oral calcium  fosfomycin: are urine levels sufficient to ensure efficacy for urinary tract infections?" is given below calcium fosfomycin, calcium can affect the efficiency of medicine  or important for stability of antibiotic?

No evidence of calcium effect on efficacy and stability has been described.

Abstract: This section is okay but in the last  conclusion, urine concentrations and PK/PD analysis and Monte Carlo simulations support calcium fosfomycin for the treatment of UTIs in women. is looking incomplete, author should provide how much amount of dose is efficient or significant for patient .

We thank the reviewer for this comment. We have modified the conclusion of the abstract. In the new version of the manuscript, it is as follows:

Our results show that oral calcium fosfomycin at a dose level of 1000 mg every 8 h provides urine concentrations sufficient to ensure efficacy for the treatment of UTIs in women.

 Introduction: 

The dose in adults is 500 mg-1000 mg every 8 hours. Up to date, the available information and a positive safety profile of this drug are considered enough to stablish a positive benefit- risk balance for this indication (reference required).

The following reference has been added (ref. 14):

European Medicine Agency. Fosfomycin-containing medicinal products. Fosfomycin Article 31- Referral-Annex II. 2020. Available at https://www.ema.europa.eu/en/medicines/human/referrals/fosfomycin-containing-medicinal-products#all-documents-section. Last accessed 19/03/2023.

Results and Discussion; Both Section Okay

We thank the reviewer for this comment very much.

Image: Author should remove the inside lines of all graph make more clear image  of all image 

We than this comment of the reviewer. However, we consider that inner lines facilitate the reading of data in the graph.

Conclusion: Author should more elaborate it 

We thank the reviewer for this comment. We have modified the conclusion section. In the new version of the manuscript, it is as follows:

Our study confirms the utility of PK/PD and Monte Carlo simulations to predict the efficacy of antibiotics excreted in urine and used for the treatment of UTI. One advantage of this kind of studies is the possibility of comparing different dose regimens.

The PK/PD breakpoints of calcium fosfomycin estimated from urine pharmacokinetic data resulted to be 8 mg/L, 16 mg/L and 32 mg/L for single dose of 500 mg, single dose of 1000 mg, and 3 g daily for three days, respectively. Only for the lowest dose level, the PK/PD breakpoint matches the clinical breakpoint of EUCAST for E. coli. For empiric treatment, and considering the MIC distribution reported by EUCAST, calcium fosfomycin provides a high (>95%) probability of treatment success.

Our results show that oral calcium fosfomycin at a dose level of 1000 mg every 8 h for three days provides urine concentrations sufficient to ensure efficacy for the treatment of UTIs in women. However, the results obtained in this study carried out with healthy women and based on urine PK/PD analysis must be correlated with well-designed efficacy studies in women with UTIs.

Round 2

Reviewer 2 Report

In general, questions were solved, and improvements were noted.

Regarding Question Q5, my suggestion is to include Table 1 as supplementary data.

Regarding Question Q7, english should be improved.

Lines 222-223: please confirm data values regarding “respectively” order (cut-off vs clinical breakpoint).

Corrections to the “Data Availability Statement” are needed.

Author Response

Regarding Question Q5, my suggestion is to include Table 1 as supplementary data.

Table 1 has moved to Supplementary material. The other tables have been renumbered.

Regarding Question Q7, English should be improved.

English has been revised.

Lines 222-223: please confirm data values regarding “respectively” order (cut-off vs clinical breakpoint).

Thank for this comment. It is a mistake and we have corrected it.

Corrections to the “Data Availability Statement” are needed.

This has been corrected